# Sobrerol: New Perspectives to Manage Patients with Frequent Respiratory Infections

**DOI:** 10.3390/children10071210

**Published:** 2023-07-12

**Authors:** Giorgio Ciprandi, Attilio Varriccchio

**Affiliations:** 1Allergy Center, Casa di Cura Villa Montallegro, 16145 Genoa, Italy; 2Department of Otolaryngology, University of Molise, 86100 Campobasso, Italy; attilio.varricchio@aivas.it

**Keywords:** mucus, sobrerol, mucolytic agent, respiratory tract infections, mechanisms of action, antioxidant activity, safety

## Abstract

Respiratory tract infections (RTIs) are usually characterized by mucus hypersecretion. This condition may worsen and prolong symptoms and signs. For this reason, reducing mucus production and improving mucus removal represent relevant aspects of managing patients with RTIs. In this regard, mucoactive drugs may be effective. Mucoactive agents constitute a large class of compounds characterized by different mechanisms of action. Sobrerol is a monoterpene able to fluidify mucus, increase mucociliary clearance, and exert antioxidant activity. Sobrerol is available in various formulations (granules, syrup, nebulized, and suppository). Sobrerol has been on the market for over 50 years. Therefore, the present article revised the evidence concerning this compound and proposed new possible strategies. The literature analysis showed that several studies investigated the efficacy and safety of sobrerol in acute and chronic RTIs characterized by mucus hyperproduction. Seven pediatric studies have been conducted with favorable outcomes. However, the regulatory agencies recently reduced the treatment duration to three days. Therefore, a future study will test the hypothesis that a combination of oral and topical sobrerol could benefit children and adults with frequent respiratory tract infections. The rationale of this new approach is based on the concept that mucus accumulation could be a risk factor for increased susceptibility to infections.

## 1. Background on Respiratory Tract Infections

Respiratory tract infections (RTIs) are widespread diseases and constitute among the most frequent causes of access to primary care doctors, as recently reported by a systematic review including data from 12 countries across five continents [1]. Acute upper RTIs are prevalent and usually recognize a bacterial or viral cause, although viral forms are the most common, as reported in the literature [2]. It has to be underlined that, in clinical practice, the diagnosis is given without a known pathogen. Many viral agents may cause acute upper RTIs, but rhinovirus, coronavirus, syncytial, influenza, parainfluenza, adenovirus, coxsackievirus, echovirus, paramyxovirus, and enterovirus are the most common [3,4,5,6,7,8]. Viral respiratory infections are typically seasonal, such as increased infectivity during cold seasons [9]. Exposure to cold exerts different mechanisms to promote RTIs, including impaired mucociliary clearance, deficient nasal defense, and reduced immune function [10]. Clinically, we have to consider that all respiratory viruses may induce an influenza-like illness (ILI), also named flu-like syndrome [11]. Indeed, ILI represents a typical acute viral illness and mimics the clinical features of influenza [12]. ILI is also an acute medical condition characterized by general and respiratory symptoms. In particular, the definition of ILI (common throughout Europe) includes “any person who presents a sudden and rapid onset of at least one of the following general symptoms: fever or feverishness, malaise/exhaustion, headache, myalgia, and at least one of the following respiratory symptoms: cough, sore throat, and wheezing” [13]. However, if milder manifestations occur, the typical common cold has to be considered [14]. Even if AURI may be associated with lower respiratory tract involvement, a healthy and immunocompetent subject usually presents only symptoms concerning the upper respiratory tract. We also must remember that common cold and ILI are umbrella terms that include different conditions, including rhinitis, sinusitis, pharyngitis, laryngitis, otitis, and tonsillitis [15].

These RTIs, mainly in children and adolescents, have a significant economic burden on the family and society.

## 2. Practical Management

From a clinical point of view, AURI management is generally grounded in a quick treatment that presumes a diagnosis usually based on clinical and epidemiological criteria [16]. The most common symptoms include sneezing, rhinorrhea, nasal congestion, hypo/anosmia, hypo/ageusia, facial pressure, sore throat, cough, headache, discomfort, myalgias, and low-grade fever [17]. Notably, these symptoms usually last less than ten days, apart from the cough, which tends to last longer [18]. The treatment should be timely and appropriate for every single patient. However, if symptoms persist longer or worsen, a trivial common cold may evolve into rhinosinusitis, needing an appropriate work-up [19]. As mentioned above, it has to be underlined that the cough may even last for more than a month in some subjects [20]. Another aspect that has to be considered is that even if these symptoms are self-resolving, they are still particularly annoying. The parents want to solve them immediately, mainly if a fever is present. Fever often instills fear in parents, even unmotivated fear, so a real fever-phobia is generated [21].

Consequently, doctors prescribe symptomatic relievers as a first-line treatment [22]. The main goal of treatment is, in fact, the prompt reduction of symptom intensity and duration. It is vital to recommend to patients and parents that antibiotics should not be used unless a bacterial complication occurs [23]. Non-steroidal anti-inflammatory drugs (NSAIDs), nasal lavage, and non-pharmacologic remedies are usually sufficient to rapidly cure the most acute viral infections [24]. Another aspect that we still need to consider has been further emphasized by the recent COVID-19 pandemic [25]. We should believe that facing an infection is always accompanied by inflammation [26]. As a result, inflammation dampening represents a leading therapeutic target in managing infections. The second lesson provided by COVID-19 infection concerns the dramatic outbreak of other RTIs that occurred successively. The explanation of this infections epidemy depends on the restrictive measure (lockdown, mask use, and social distancing) that significantly diminished the incidence of RTIs [27]. The paradigmatic example was the negligible prevalence of bronchiolitis during the early COVID-19 pandemic [28]. However, since the slackening of restrictive measures, there has been a surge in cases of bronchiolitis that has put a strain on the pediatric hospital network [29]. The 2022/2023 seasonal influenza epidemic had an early onset, extraordinary incidence, clinical severity, and persistent duration of the epidemic plateau [30]. Other RTIs, mostly viral, spread at the same time [31]. Therefore, RTIs have become increasingly impactful in daily life. A poorly trained immune system contributed significantly to this increased infection susceptibility. COVID-19 is also known to cause high rates of extra-pulmonary disease concomitant with the pulmonary syndrome that should be noted [32]. The high rate of these concomitant diseases accounts for the difficulty and delay in their diagnosis. Consequently, careful management of RTIs is required today, considering all factors that could contribute to worsening the infection evolution.

## 3. The Relevance of Mucus Hyperproduction in Respiratory Tract Infections

The respiratory mucus secretion is a complex mixture produced by different structures, including submucosal glands and secreting epithelial cells (goblet cells and Clara cells) [33,34]. The submucosal glands are tubular/tubulacinar formations producing mucous, serous, or seromucous secretions. The goblets cells are mucous-secreting cells intercalated in the cylindrical ciliated epithelium lining, disappearing in the terminal bronchioles. The Clara cells are intercalated with the low or cubic cylindrical epithelium, ciliated or not, which extends in a single layer along the peripheral airways. These cells produce both a mucous secretion (like goblet cells) and a lipoprotein secretion (like type B pneumocytes), which can be identified as an alveolar surfactant [35,36].

The amount of mucus produced depends on the number of mucus-secreting cells present at that level, which in turn is related to the total surface area of the airways; thus, mucus production occurs more in the peripheral airways than in the central airways [37]. Physiologically, an adult produces about 10–100 mL of mucus per day, and the amount of mucus that reaches the trachea is approximately 10–20 mL/day [33,34]. The superficial layer constitutes the ‘sol’ phase of the mucus, very rich in water, about 3 microns thick, and occupies almost the entire length of the cilia of the epithelial cells [38]. Above, there is a dense layer: the ‘gel’ phase. It mainly contains glycoproteins (mucins), characterized by a central protein structure anchored to lateral polysaccharide chains formed by sialic acid (sialomucins) or fucose (fucosomucins). In the gel layer, molecules with anti-infective activity exist, such as secretory IgA (S-IgA), lactoferrin, and lysozyme. In addition, it has been hypothesized that a third layer consisting of surfactant exists between the sol and gel layers [34]. Usually, only the gel layer is transported, but the sol layer seems essential for mucus transport because it allows the cilia to beat effectively [36]. Mucus transport is governed by the mechanical forces of ciliary beating and airflow, counteracted by the friction and inertia of the mucus itself [39]. Mucociliary clearance is prevalent in the peripheral airways. Each ciliary cell has about 200 cilia, and the cilia are equipped with ‘claws’ that reach into the gel layer and push it toward the oropharynx [37,38]. The mucociliary clearance efficacy depends on the airflow velocity, a function of the airway diameter, and the pressure the expiratory muscles create [40]. Moreover, in the first years of life, the small airways tend to collapse during normal breathing, and the ciliary machinery develops progressively [41].

Mucus hypersecretion is a common manifestation shared by many pathological conditions of both the upper airways (rhinitis, rhinosinusitis, pharyngitis, laryngitis) and the lower airways (acute and chronic bronchitis, bronchiectasis, and pneumonia) [41].

Initially, the hypersecretion of mucus represents a defense system: the increased thickness of the secretion promotes the clearance and dilution of soluble particles. The inflammatory vasodilatation increases the secretory activity of mucous cells and the production of immunoglobulin and complement fractions, enhancing the function of S-IgA. The quantitative increase in mucous secretion may be associated with changed viscoelastic characteristics of the mucus, resulting from the prevalence of neutral mucins (fucosomucins) over acidic mucins (sialomucins and sulphomucins). From a rheological point of view, neutral mucins are denser and more adhesive. Later, when hypersecretion becomes excessive, the defensive role diminishes and can worsen RTIs because of impaired mucociliary clearance, wall edema, and bronchial obstruction [42].

Moreover, it is also important to note that the mucus assessment has a role in diagnosing some respiratory diseases. Mucus investigation is often used to identify the pathogens in pulmonary infections. Namely, sputum analysis is part of the routine evaluations of patients with bronchiectasis, and it was also found to have diagnostic value for patients with inflammatory lung diseases—especially among active smokers, as recently demonstrated by induced sputum [43].

In addition, chronic respiratory tract inflammation is associated with abundant secretions that are only partially reabsorbed in the airways. Hypertrophy and hyperplasia of the secretory structures contribute to maintaining hypersecretion. In addition, vasodilatation and mucosal edema cause fluid supply into mucous glands, worsening hypersecretion [44]. In other words, a vicious circle starts and self-maintains: mucus hypersecretion fosters further hypersecretion. As a result, to stop this negative loop, “mucoactive” drugs may be helpful in clinical practice.

## 4. Mucoactive Drugs

Mucoactive drugs are molecules able to modify viscous-elastic characteristics of mucus, promoting its clearance. This definition disregards the specific mechanism of action. In particular, mucolytic medications can reduce mucus viscosity by depolymerizing mucin glycoproteins. Mucolytics may be classified into two main groups: mucolytics with direct action and indirect action. Table 1 summarizes the classification of mucolytic agents based on the mechanism of action.

In clinical practice, mucoactive drugs may be a therapeutic option in medical conditions characterized by mucus hypersecretion [45]. In particular, mucolytics facilitate airway clearance in specific diseases, including bronchiectasis, bronchitis, cystic fibrosis, and rhinosinusitis, with abundant mucus production [46]. Mucolytics may relieve acute productive (wet) cough [46]. Several clinical studies investigated the usefulness of mucolytics in clinical practice. However, the evidence level generally was weak, mainly for methodological issues [47]. There is evidence that mucolytic drugs are effective in infectious and non-infectious diseases. Indeed, some mucolytic drugs may exert additional activities that may be fruitfully used in non-infectious diseases. For example, ambroxol inactivates transcription factors and suppresses pro-inflammatory cytokines; thus, it has been prospected as a new treatment option for treating ulcerative colitis [48]. N-acetylcysteine exerts antiviral activity, potentially helpful in managing patients with human immunodeficiency virus (HIV) [49]. On the other hand, it is well known that mucoactive drugs are effective in treating respiratory tract infections, mainly in patients with cystic fibrosis, bronchiectasis, and chronic bronchitis [50].

Mucolytics are commonly used in clinical practice, preferring consolidated molecules.

## 5. Sobrerol

Sobrerol has been on the market in many European countries for over 50 years since its launch in the early 1970s. Sobrerol (5-hydroxy-α, α, 4-trimethyl-3-cyclohexene-1-methanol) is a monocyclic monoterpene with two hydroxyl functions. Various effects characterize it (Figure 1). We focused on sobrerol as it is commonly used, and a large body of evidence exists on this compound. However, most studies are old, and we would propose new possible strategies for their use.

Sobrerol in vivo increased mucus production and ciliary motility, thus improving mucociliary clearance [45]. In addition, sobrerol reduced the viscosity of tracheobronchial mucus without causing any alterations of the alveolar surfactants [46]. Radical scavenging activities have also been reported [51]. Finally, sobrerol may increase the production of secretory IgAs. Sobrerol is available in different formulations, including syrup, water-soluble sachets, nebulization, intramuscular (or intravenous) vials, and suppositories. The main indication is the treatment of respiratory disorders characterized by thick, viscous hypersecretion. The recommended oral dose in adults is 600 mg (equivalent to the contents of two sachets) per day for up to three days. In children, this dosage is halved. The dose administered by the aerosol route is one vial for nebulization, containing 40 mg of sobrerol, one to two times daily. Clinical studies in the literature have used a treatment duration of up to three months or a maximum daily dosage of 900 mg for ten consecutive days [45]. The only contraindication is used in children under 30 months of age or with a history of epilepsy or febrile convulsions, as well as hypersensitivity to the active ingredient or any excipients used in the various formulations. Particular precaution must, however, be observed in subjects with severe respiratory insufficiency, asthmatics, and debilitated patients, as the increased fluidity of secretions requires effective expectoration.

As sobrerol has been prescribed for over 50 years, numerous clinical studies have been conducted, especially in the 1970s and 1980s [52]. A recent review updated and synthetically reported some of them [45]. Globally, 25 studies were published, including ten double-blind, randomized, and controlled trials and 15 open studies (controlled or non-controlled) concerning acute and chronic respiratory diseases in children and adults [51,53,54,55,56,57,58,59,60,61,62,63,64,65,66,67,68,69,70,71,72,73,74,75,76,77,78]. In particular, there are seven published pediatric studies (Table 2) regarding different medical conditions; three were randomized controlled trials (RCTs), and four were open studies [54,63,64,65,73,76,78].

The first pediatric RCT study was performed in 1981. This RCT study was conducted as double-blind, randomized, and placebo-controlled and had 100 patients aged between 12 and 74 years with acute or chronic upper or lower respiratory tract infections [54]. Sobrerol syrup (260 mg) was administered with carbocysteine capsules (375 mg) four times daily for 21 days. This combination significantly improved objective and subjective clinical parameters and lung function compared to a placebo. In addition, the treatment was well tolerated. Unfortunately, pediatric outcomes could not be deduced. The second study evaluated the efficacy and safety of oral sobrerol compared to oral N-acetylcysteine in 40 children with acute respiratory diseases (bronchitis or pharyngo-tracheobronchitis) [63]. Sobrerol was administered as granules in one-dose sachets for three days at 100 mg/3 times a day. N-acetylcysteine granules were used at 300 mg/day for three days. Clinical parameters and biological data, including rheological examination of expectorate, were considered at baseline and the end of the course. The two treatments were effective without significant differences. However, sobrerol induced a better reduction of expectorating viscosity and was better tolerated than N-acetylcysteine. A third RCT study was conducted as double-blind, randomized, and placebo-controlled in 30 children with pertussis aged between 10 months and 12 years [64]. The measured outcomes included clinical and functional parameters. The treatment consisted of an oral combination of clofedanol (central antitussive agent) 1.62 mg/Kg/daily with sobrerol 3.6 mg/Kg/day for 15 days. The active treatment was safe and significantly improved signs and lung function compared to placebo. The first open study included 40 children under five years old with acute and recurrent bronchitis [65]. This study randomly compared oral sobrerol (50–100 mg/twice daily) with oral bromhexine (2–4 mg/three times daily) for two weeks. The outcome was an improvement rate. The results showed that both drugs were effective, but no significant differences occurred. The second open study retrospectively recruited 59 children with acute upper RTIs and wet cough aged between 3 and 14 years [73]. This study compared children treated with oral antibiotics (amoxicillin or a macrolide) with children treated with nebulized mucoactive drugs (sobrerol or N-acetylcysteine) used in standard practice. The children treated with mucolytics significantly improved clinical parameters compared with children treated with antibiotics. A third open study considered a group of 30 children (5–10 years old) with secretory otitis media and treated with nebulized sobrerol (one 40 mg vial/day) alone for ten consecutive days [76]. The treatment improved clinical outcomes (nasal obstruction, deafness, and earache) and impedance values. The improvement depended on mucus fluidification in the upper airways. The previous observational study included 20 children (6 months–2 years old) with pertussis [78]. The primary outcome was the time course to symptom resolution. Children were treated with oral or rectal sobrerol (four times daily), oral salbutamol, and oral erythromycin (40 mg/kg/day) until clinical resolution. This treatment was considered better than historical controls (antibiotics alone or associated with hyperimmune gamma globulins and/or cough sedatives).

Notably, all these studies were conducted in Italy. The global outcomes were positive; however, the records were heterogeneous concerning the medical conditions and treatments. The clinical judgment was always favorable, as confirmed by the consolidated use in clinical practice. However, regulatory agencies recently restricted the treatment duration to only three days for all formulations. This limitation may prompt new possible strategies for using sobrerol.

## 6. Future Perspectives

Acute RTIs significantly burden the healthcare system and single subjects, mainly in childhood. In particular, a relevant quote about toddlers having frequent RTIs. Children frequently contract RTIs because their immune systems are still partially immature. In addition, several risk factors may be involved in increasing the susceptibility to have frequent RTIs, including age (the smaller they are, the sicker they become), preschool attendance, indoor and outdoor pollution, passive smoking, poor socioeconomic status, and allergic diseases [79]. As a consequence, managing children with RIs represents a crucial task in clinical practice. RTIs significantly impact morbidity, healthcare costs, overmedication (mainly concerning antibiotics), and family quality of life. Antipyretic and antibiotic treatment is common, but their use is often empiric and not based on critical reasoning. Overuse of antibiotics accounts for the overwhelming problem of bacterial resistance, and abuse of antipyretics also exposes to the risk of adverse events. Therefore, preventive strategies are welcome. In this regard, an attractive idea could be to use a combination of two sobrerol formulations, such as oral (syrup) and nebulization (vial), during RTIs. The rationale of this combination is to achieve an optimal effect respecting the short duration (three days). The use of mucolytics usually lasts for 1–2 weeks or at least until clinical resolution. The combined use of topical and systemic routes could assure an ideal fluidification of secretions that could theoretically reduce some factors involved in the frequency of infections. Clean airways and restored mucociliary clearance could prevent pathogens’ colonization [80]. There is consistent agreement concerning the clinical and pathophysiological relevance of “clean” airways [81]. Open and clean airways guarantee the physiological functioning of the nose, mainly concerning the heating, humidification, and filtration of inhaled air. Therefore, the removal of excess mucus can contribute significantly to improving respiratory tract function. Another possible outcome could be the sobrerol effect on biofilms. Biofilms are bacteria aggregates living in a matrix composed of their own secretions, where bacteria replication is limited, and nutrients circulate through a three-dimensional network of internal channels [82]. Bacteria account for 5–35% of the biofilm volume, while 65–95% consists of the extracellular matrix, made of 97% of water and 2% proteins—DNA and RNA (both <1%) and 1–2% of polysaccharides [83]. Biofilms can be described as a particular form of colonization of human cavities and surfaces, where myriad microbial species are present, according to metagenomic investigations [84]. Therefore, biofilm represents a practical and efficient mode for bacteria ensuring remarkable survival and resistance against aggressive mechanisms, including antibiotics. As a result, being able to intact the biofilm capsule and thus disrupt it might be a desirable effect for a mucoactive drug. In this regard, it has been demonstrated that ambroxol, a mucolytic agent, could disrupt biofilms [85]. Another relevant aspect is the adjunctive use of mucoactive agents in multidrug-resistant infections. A recent review investigated the evidence on the use of a well-known mucolytic drug (NM-acetylcysteine) with particular attention to this issue [86]. Bacterial antibiotic resistance represents an alarming problem for healthcare systems in recent decades, as it has caused several serious infections, also lethal [87]. The global burden of this issue accounted for 4.95 million deaths in 2019 [88]. This dramatic fact underlines the seriousness of this phenomenon. A possible explanation of this severity is that infections caused by multidrug-resistant (MDR) bacteria are generally associated with a poor prognosis and more than 40% mortality, especially in the presence of septic shock [89]. Biofilms raise the resistance to host defenses and diminish susceptibility to antimicrobial agents so making persistent infections difficult or impossible for the immune system to clear and be eradicated with antibiotics [90]. These aspects further underscore the relevance of disrupting biofilms by mucoactive agents. The current literature provided evidence that N-acetylcysteine could actively affect biofilms [86]. Consequently, further investigation should evaluate whether sobrerol can also affect biofilm and potentially reduce the impact of multidrug-resistant pathogens.

There is also the need to deepen and update knowledge about this molecule. Further extensive prospective randomized controlled studies are needed to compare sobrerol with other agents and, more important, show evidence of clinical outcomes, such as disease duration, prevention of complications, and long-term outcomes.

## 7. Conclusive Remarks

Respiratory tract infections are a common medical condition affecting virtually the entire population. In addition, RTIs significantly affect the healthcare system and patients’ quality of life. Mucus hypersecretion is a typical consequence of infection and could promote recurrence or prolong infection duration. Therefore, mucus fluidification could represent a therapeutical option currently used in clinical practice. In this regard, using a mucolytic (sobrerol) using two administration routes could define an innovative strategy regarding the requested posology.

## Figures and Tables

**Figure 1 children-10-01210-f001:**
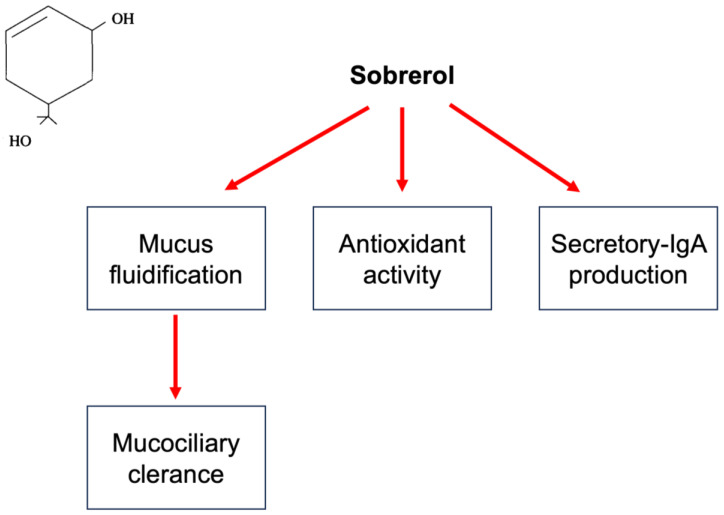
Mechanisms of action of sobrerol. In the left quadrant is reported the chemical molecule. Sobrerol, as schematized, induces mucus fluidification and consequently improves mucociliary clearance, exerts antioxidant activity, and increases sIgA production.

**Table 1 children-10-01210-t001:** Classification of mucolytic agents based on specific mechanisms of action.

Direct Action			Indirect Action					
Drugs depolymerizing mucins			Drugs modifying mucus secretion	Drugs modifying adhesivity of gel layer	Drugs modifying the sol layer	Volatile and balsamic agents	Drugs stimulating gastric reflex (cough inducing)	Drugs modifying secretion
Thiolics	Enzymes	Other molecules	S-carbossimethylcystein Sobrerol Domiodol Hydropropylidenglycerol	Ambroxol Bromexine Propylene-glycol Etasulphate sodic Bicarbonate sodic	H_2_O Potassium salts Sodium salts	Pinanes Terpenes Methanes Phenol derivates	Ammonium chloride Sodium citrate Guaifenesin Ipecap	Drenergic agents Cholinergic agents Corticosteroid Antihistamines
Cisteine Methylciytein Ethylciytein Acetylcysteine Erdostein Ludostein Stepronin Thipronin Mesna	Trypsin Streptokinase Serratiopeptidase Stericase	Urea Ascorbic acid Hypertonic saline Inorganic iodures						

**Table 2 children-10-01210-t002:** Pediatric studies using sobrerol.

Authors, Year (Ref)	Study Design	Disease	Patients Number (Age)	Primary Outcomes	Treatments (Dosage and Duration)	Results
Milvio et al., 1981 [54]	RCT	Acute and chronic infections	50/50 (12–74 years)	Lung function Expectorate	Oral sobrerol (260 mg) + carbocysteine (375 mg) × 4/day Placebo 14–21 days	Active treatment significantly improved all outcomes
Seidita et al., 1984 [63]	RCT	Acute respiratory infections	20/20 (3–12 years)	Clinical signs Expectorate Biological data	Oral sobrerol (100 mg × 3/day) Oral N-Acteylcysteine (100 mg × 3/day) One week	Sobrerol significantly reduced mucus viscosity
Miraglia del Giudice et al., 1984 [64]	RCT	Pertussis	15/15 (10 months–12 years)	Clinical signs Lung function	Clofedanol (1.62 mg/kg/d) + oral sobrerol (6 mg/kg/d) Placebo 15 days	Active treatment improved parameters more quickly
Azzolini et al., 1990 [65]	Open	Acute and recurrent bronchitis	40 (<5 years)	Improvement rate	Oral sobrerol (50–100 mg × 2/d) Oral brohexine (2–4 × 3/d) 2 weeks	No difference
Zanasi et al., 2012 [73]	Open	Acute upper respiratory tract infections with wet cough	59 (3–14 years)	Severity, frequency, and duration of cough	Antibiotics (amoxicillin or erythromycic) Nebulized sobrerol or N-Acetylcysteine.	Mucolytics improved cough
Bellussi et al., 1981 [76]	Open	Secretory otitis media	30 (5–10 years)	Nasal obstruction, earache, deafness	Nbulized sobrerol (40 mg/d) 10 days	Significant reduction of all outcomes
Crosca et al., 1982 [78]	Open	Pertussis	20 (6 months–2 years)	Time to resolution	Oral or rectal sobrerol Salbutamol Erythromycin Until resolution	Better outcomes in comparison with historical records

## Data Availability

Data are present in the literature.

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
