# Peer review of "Sobrerol: New Perspectives to Manage Patients with Frequent Respiratory Infections"

_children, 2023, doi:10.3390/children10071210_

Round 1

Reviewer 1 Report

I would like to thank you for the opportunity to review this manuscript. The authors describe the current data on the effectiveness of Sobrerol for pediatric patients with acute URI and address URI and the physiology of mucus secretion in general. This paper is very interesting and well-written. Several issues should be addressed by the authors:

1.      General grammar issues, such as: "the clinical feature of influenza" should be features (line 36), lines 47-48 should be rephrased, "commonly" (line) is not needed, etc.

2.      Background on respiratory infections:

a.      " and usually recognize a bacterial or viral cause" – is it correct that in most cases the pathogen is found? Most time a clinical diagnosis is given without a known pathogen. Please address this issue and correct if needed.

3.      Practical management: This section is well-written and informative. I think the authors did a great job by speaking on the COVID-19 itself and its effects on other viral infections. In this regard, COVID-19 is also known to cause high rates of extra-pulmonary disease concomitant with the pulmonary syndrome that should be noted. For this I recommend the authors to use the following paper which address the high rate of concomitant disease and the difficulty in its diagnosis:

https://pubmed.ncbi.nlm.nih.gov/36779316/

4.      The relevance of mucus hyper production: Once again this part is comprehensive and the authors did a good job in describing the rationale for mucoactive agents and the general physiology of mucus production. I recommend the authors to add data on the role of mucus in diagnosis of diseases – It is often used to identify the pathogens in pulmonary infections, it is part of the routine evaluations of patients with bronchiectasis, and it was also found to have diagnostic value for patients with inflammatory lung diseases – especially among active smokers. For this I recommend the authors to use the following paper which examined this field and found the value of sputum particles in diagnosis - https://pubmed.ncbi.nlm.nih.gov/36975802/

5.      Muco-active drugs:

a.      Why in Table 1 there are three columns used for the direct action drugs and only two for the rest? Please also add headings to each column.

b.      Please add additional information on previous studies that investigated the utility of mucolytic drugs. The authors say that the evidence level is weak (which I agree) but citations should be added to support their notion. In addition, I suggest adding examples on their efficacy in infectious and non-infectious diseases.

6.      Sobrerol:

a.      I think the authors should add why did they focused on sobrerol compared with other agents? Is it because of its safety? Common use? Large body of evidence?

7.      Future perspectives: I recommend the authors to add their perspective on the main aim of future studies in this field. I believe that further large prospective randomized studies are needed to compare Sobrerol with other agents and more important, show evidence on clinical outcomes, such as disease duration, prevention of complications and long-term outcomes.  

See my note from above.

Author Response

First of all, we would thank the Reviewer for the helpful comments.

I would like to thank you for the opportunity to review this manuscript. The authors describe the current data on the effectiveness of Sobrerol for pediatric patients with acute URI and address URI and the physiology of mucus secretion in general. This paper is very interesting and well-written. Several issues should be addressed by the authors:

  1. General grammar issues, such as: "the clinical feature of influenza" should be features (line 36), lines 47-48 should be rephrased, "commonly" (line) is not needed, etc.

R Many thanks for this comment. We amended these errors.

  1. Background on respiratory infections:
  2. " and usually recognize a bacterial or viral cause" – is it correct that in most cases the pathogen is found? Most time a clinical diagnosis is given without a known pathogen. Please address this issue and correct if needed.

R Many thanks for this comment. We rephrased the sentence accordingly.

  1. Practical management: This section is well-written and informative. I think the authors did a great job by speaking on the COVID-19 itself and its effects on other viral infections. In this regard, COVID-19 is also known to cause high rates of extra-pulmonary disease concomitant with the pulmonary syndrome that should be noted. For this I recommend the authors to use the following paper which address the high rate of concomitant disease and the difficulty in its diagnosis:

https://pubmed.ncbi.nlm.nih.gov/36779316/

R Many thanks for this comment. We implemented the text accordingly and cited this reference.

  1. The relevance of mucus hyper production: Once again this part is comprehensive and the authors did a good job in describing the rationale for mucoactive agents and the general physiology of mucus production. I recommend the authors to add data on the role of mucus in diagnosis of diseases – It is often used to identify the pathogens in pulmonary infections, it is part of the routine evaluations of patients with bronchiectasis, and it was also found to have diagnostic value for patients with inflammatory lung diseases – especially among active smokers. For this I recommend the authors to use the following paper which examined this field and found the value of sputum particles in diagnosis - https://pubmed.ncbi.nlm.nih.gov/36975802/

     R Many thanks for this comment. We implemented the text accordingly and cited this reference.

  1. Muco-active drugs:
  2. Why in Table 1 there are three columns used for the direct action drugs and only two for the rest? Please also add headings to each column.

R Many thanks for this comment. We revised the table accordingly.

  1. Please add additional information on previous studies that investigated the utility of mucolytic drugs. The authors say that the evidence level is weak (which I agree) but citations should be added to support their notion. In addition, I suggest adding examples on their efficacy in infectious and non-infectious diseases.

R Many thanks for this comment. We added the reference. We also provided some synthetic examples as requested.

  1. Sobrerol:
  2. I think the authors should add why did they focused on sobrerol compared with other agents? Is it because of its safety? Common use? Large body of evidence?

R Many thanks for this comment. We provided the reason concerning the focusing on sobrerol.

  1. Future perspectives: I recommend the authors to add their perspective on the main aim of future studies in this field. I believe that further large prospective randomized studies are needed to compare Sobrerol with other agents and more important, show evidence on clinical outcomes, such as disease duration, prevention of complications and long-term outcomes.  

R Many thanks for this comment. We expanded the text including these interesting suggestions.

Reviewer 2 Report

- The last sentence in abstract is not in the right place.

- The idea of bronchiolitis (kine started from 77) is not clear and not enough.

- What is the reference for Table 1? please, mention in the manuscript.

- Usually the title of the figure is under the figure.

- There are some extra spaces in the file (design issue).

- Line 197-199: add the year.

- Add the publication year to Table 2.  

Need minor modifications

Author Response

First of all, we would thank the Reviewer for the helpful comments.

- The last sentence in abstract is not in the right place.

R Many thanks for this comment. We amended it.

- The idea of bronchiolitis (kine started from 77) is not clear and not enough.

R Many thanks for this comment. Actually, bronchiolitis has been envisaged as a typical model of increased susceptibility consequent to COVID-19.

- What is the reference for Table 1? please, mention in the manuscript.

R Many thanks for this comment. However, we provided the mention in the text (present line 157).

- Usually the title of the figure is under the figure.

R Many thanks for this comment. We moved it.

- There are some extra spaces in the file (design issue).

R Many thanks for this comment. Amended

- Line 197-199: add the year.

R Many thanks for this comment. Added

- Add the publication year to Table 2.  

R Many thanks for this comment. Added

Reviewer 3 Report

Dear Authors,

Your manuscript has been reviewed,

This paper deserves attention since it highlights a very important topic,

Kindly find below my remarks (major and minor ones):

01- In the Line 1, Authors are invited to replace "Type of the Paper Perspective" by "Perspective".

02- The Abstract section is weak and not very clear for readers, objective and results are not well represented.

03- In the Whole Manuscript, Authors are invited to replace "Respiratory infections" by "Respiratory Tract Infections", they are also invited to use the Abbreviation "RTIs".

04- In Lines 10-11, this sentence is not clear at all " Mucoactive agents include different molecules with different mechanisms of action", Authors are invited to rewrite it in a better way.

05- Keywords (in Line 20) are very weak, authors are invited to change and to add more keywords.

06- In Line 67, Authors are invited to explain what do they mean by "NSAIDs"?

07- In Lines 68-70, after this sentence "Another aspect that we still need to consider has been further emphasized by the recent COVID-19 pandemic." Authors are invited to add the following reference:

REF 1: Risk Markers of COVID-19, a Study from South-Lebanon

08- In Lines 72-76, after these two sentences "The second lesson provided by COVID-19 infection concerns the dramatic outbreak of other respiratory infections that occurred successively. The explanation of this infections epidemy depends on the restrictive measure (lockdown, mask use, and social distancing) that significantly diminished the incidence of respiratory infections." Authors are invited to add the following reference:

REF 2: The emergence of SARS-CoV-2 variant (s) and its impact on the prevalence of COVID-19 cases in the Nabatieh Region, Lebanon

09- In Lines 91-97, after these sentences "The submucosal glands are tubular/tubulacinar formations producing mucous, serous, or seromucous secretions. The goblets cells are mucous-secreting cells intercalated in the cylindrical ciliated epithelium lining, disappearing in the terminal bronchioles. The Clara cells are intercalated with the low or cubic cylindrical epithelium, ciliated or not, which extends in a single layer along the peripheral airways. These cells produce both a mucous secretion (like goblet cells) and a lipoprotein secretion (like type B pneumocytes), which can be identified as an alveolar surfactant." Authors are invited to add references related to these sentences such as the follow:

REF 3: Morpho-functional characterization of the submucosal glands at the nasopharyngeal end of the auditory tube in humans

REF 4: New developments in goblet cell mucus secretion and function

REF 5: Clara cell: progenitor for the bronchiolar epithelium

10- Table 1 is not clear at all, third and fourth columns are not clear, we do not know what they are informing readers about!! Also it is very large (3 pages).

11- Figure 1 is not clear for readers, also Figures' legends are always below figures. 

12- In the whole manuscript, some lines are missed (they are empty) example: Lines 184-186 and 194-196, etc.

13- In the Line 193, Authors are invited to explain what do they mean by RCT? Is it Research Clinical Trial? Please explain.

Best Regards,

Dear Authors,

Kindly find my remarks concerning the English language in the manuscript:

01- Some sentences are not clear.

02- Some sentences are very short.

03- Some English polishing is needed in the manuscript.

Author Response

First of all, we would thank the Reviewer for the helpful comments.

Dear Authors,

Your manuscript has been reviewed,

This paper deserves attention since it highlights a very important topic,

Kindly find below my remarks (major and minor ones):

01- In the Line 1, Authors are invited to replace "Type of the Paper Perspective" by "Perspective".

R Many thanks for this comment. Amended

02- The Abstract section is weak and not very clear for readers, objective and results are not well represented.

R Many thanks for this comment.  We revised the text accordingly.

03- In the Whole Manuscript, Authors are invited to replace "Respiratory infections" by "Respiratory Tract Infections", they are also invited to use the Abbreviation "RTIs".

R Many thanks for this comment.  We revised the text accordingly.

04- In Lines 10-11, this sentence is not clear at all " Mucoactive agents include different molecules with different mechanisms of action", Authors are invited to rewrite it in a better way.

R Many thanks for this comment.  We revised the text accordingly.

05- Keywords (in Line 20) are very weak, authors are invited to change and to add more keywords.

R Many thanks for this comment.  We implemented them accordingly.

06- In Line 67, Authors are invited to explain what do they mean by "NSAIDs"?

R Many thanks for this comment.  We provided the explanation.

07- In Lines 68-70, after this sentence "Another aspect that we still need to consider has been further emphasized by the recent COVID-19 pandemic." Authors are invited to add the following reference:

REF 1: Risk Markers of COVID-19, a Study from South-Lebanon

R Many thanks for this comment.  However, we need the exact reference, as more than 2,000 studies concerned it.

08- In Lines 72-76, after these two sentences "The second lesson provided by COVID-19 infection concerns the dramatic outbreak of other respiratory infections that occurred successively. The explanation of this infections epidemy depends on the restrictive measure (lockdown, mask use, and social distancing) that significantly diminished the incidence of respiratory infections." Authors are invited to add the following reference:

REF 2: The emergence of SARS-CoV-2 variant (s) and its impact on the prevalence of COVID-19 cases in the Nabatieh Region, Lebanon

R Many thanks for this comment. However, we believe that this study does add any relevant information on the discussed issue.

09- In Lines 91-97, after these sentences "The submucosal glands are tubular/tubulacinar formations producing mucous, serous, or seromucous secretions. The goblets cells are mucous-secreting cells intercalated in the cylindrical ciliated epithelium lining, disappearing in the terminal bronchioles. The Clara cells are intercalated with the low or cubic cylindrical epithelium, ciliated or not, which extends in a single layer along the peripheral airways. These cells produce both a mucous secretion (like goblet cells) and a lipoprotein secretion (like type B pneumocytes), which can be identified as an alveolar surfactant." Authors are invited to add references related to these sentences such as the follow:

REF 3: Morpho-functional characterization of the submucosal glands at the nasopharyngeal end of the auditory tube in humans

REF 4: New developments in goblet cell mucus secretion and function

REF 5: Clara cell: progenitor for the bronchiolar epithelium

R Many thanks for this comment. We added some studies on this issue.

10- Table 1 is not clear at all, third and fourth columns are not clear, we do not know what they are informing readers about!! Also it is very large (3 pages).

R Many thanks for this comment. We provided a new table.

11- Figure 1 is not clear for readers, also Figures' legends are always below figures. 

R Many thanks for this comment. We moved the legend and implemented it.

12- In the whole manuscript, some lines are missed (they are empty) example: Lines 184-186 and 194-196, etc.

R Many thanks for this comment. We amended it.

13- In the Line 193, Authors are invited to explain what do they mean by RCT? Is it Research Clinical Trial? Please explain.

R Many thanks for this comment. Provided.

Round 2

Reviewer 1 Report

I would like to thank you for the opportunity to re-review this interesting paper. Overall, the authors addressed all the issues raised in my previous review and the manuscript has significantly improved. I have no additional major issues that the authors should address. Still, I would like to mention that there remain some minor grammar issues (such as repetitive connective words between sentences) and I recommend the authors in their future responses to reviewers to add the specific lines in which changes were made. Thank you and good luck

See above

Author Response

I would like to thank you for the opportunity to re-review this interesting paper. Overall, the authors addressed all the issues raised in my previous review and the manuscript has significantly improved. I have no additional major issues that the authors should address. Still, I would like to mention that there remain some minor grammar issues (such as repetitive connective words between sentences) and I recommend the authors in their future responses to reviewers to add the specific lines in which changes were made. Thank you and good luck

R Many thanks for this comment and the second review of our paper. We revised the manuscript removing some connective words between sentences. In detail (as requested) lines: 32, 36, 37, 41, 49, 60, 66, 71, 74, 83, 85, 86, 87, 89, 139, 145, 163, 166, 170, 175, 188, 267, 269, 274, 278, 295.

Reviewer 3 Report

Dear Authors,

Your revised manuscript has been reviewed,

I would like to thank you for the modifications you did,

I just want to clarify some of my points:

Concerning my point number 07, the full information regarding the reference are present below:

Risk Markers of COVID-19, a Study from South-Lebanon, Chakkour et al., 2022, COVID, MDPI.

Concerning my point number 08, The mentioned article "The emergence of SARS-CoV-2 variant (s) and its impact on the prevalence of COVID-19 cases in the Nabatieh Region, Lebanon" highlights the importance of Lockdown, Social Distancing etc. on COVID-19, therefore I believe it is related to this point.

Best Regards,

Author Response

Your revised manuscript has been reviewed,

I would like to thank you for the modifications you did,

I just want to clarify some of my points:

Concerning my point number 07, the full information regarding the reference are present below:

Risk Markers of COVID-19, a Study from South-Lebanon, Chakkour et al., 2022, COVID, MDPI.

R Many thanks for this clarification. Actually, the Journal is not indexed in PubMed or Scopus. This study investigated some risk factors for COVID-19 in South-Lebanon. Sincerely, we do not see the relevance of the outcomes of this study to the sentence in our manuscript. It honestly seems to us that it is not necessary to further support the sentence. In fact, there is already a more comprehensive bibliographical reference.

Concerning my point number 08, The mentioned article "The emergence of SARS-CoV-2 variant (s) and its impact on the prevalence of COVID-19 cases in the Nabatieh Region, Lebanon" highlights the importance of Lockdown, Social Distancing etc. on COVID-19, therefore I believe it is related to this point.

R Many thanks for this comment. We added this reference in the text.